# A drug design strategy based on molecular docking and molecular dynamics simulations applied to development of inhibitor against triple-negative breast cancer by Scutellarein derivatives

**Shopnil Akash**[1]*, **Farjana Islam Aovi**[1], **Md. A. K. Azad**[1], **Ajoy Kumer**[2], **Unesco Chakma**[3], **Md. Rezaul Islam**[1], **Nobendu Mukerjee**[4,5], **Md. Mominur Rahman**[1], **Imren Bayıl**[6], **Summya Rashid**[7], **Rohit Sharma**[8]*

1 Department of Pharmacy, Faculty of Allied Health Sciences, Daffodil International University, Dhaka, Bangladesh, 2 Laboratory of Computational Research for Drug Design and Material Science, Department of Chemistry, European University of Bangladesh, Dhaka, Bangladesh, 3 School of Electronic Science and Engineering, Southeast University, Nanjing, China, 4 Center for Global Health Research, Saveetha Medical College and Hospital, Saveetha Institute of Medical and Technical Sciences, Chennai, India, 5 Department of Microbiology, West Bengal State University, West Bengal, Kolkata, India, 6 Department of Bioinformatics and Computational Biology, Gaziantep University, Gaziantep, Turkey, 7 Department of Pharmacology and Toxicology, College of Pharmacy, Prince Sattam Bin Abdulaziz University, Al-Kharj, Saudi Arabia, 8 Department of Rasa Shastra and Bhaishajya Kalpana, Faculty of Ayurveda, Institute of Medical Science, Banaras Hindu University, Varanasi, India

* shopnil.ph@gmail.com (SA); rohitsharma@bhu.ac.in (RS)

**Data Availability Statement:** All relevant data are given within the paper.

## Abstract

Triple-negative breast cancer (TNBC), accounting for 10–15% of all breast malignancies, is more prevalent in women under 40, particularly in those of African descent or carrying the BRCA1 mutation. TNBC is characterized by the absence of estrogen and progesterone receptors (ER, PR) and low or elevated HER2 expression. It represents a particularly aggressive form of breast cancer with limited therapeutic options and a poorer prognosis. In our study, we utilized the protein of TNBC collected from the Protein Data Bank (PDB) with the most stable configuration. We selected Scutellarein, a bioactive molecule renowned for its anti-cancer properties, and used its derivatives to design potential anti-cancer drugs employing computational tools. We applied and modified structural activity relationship methods to these derivatives and evaluated the probability of active (Pa) and inactive (Pi) outcomes using pass prediction scores. Furthermore, we employed in-silico approaches such as the assessment of absorption, distribution, metabolism, excretion, and toxicity (ADMET) parameters, and quantum calculations through density functional theory (DFT). Within the DFT calculations, we analyzed Frontier Molecular Orbitals, specifically the Highest Occupied Molecular Orbital (HOMO) and Lowest Unoccupied Molecular Orbital (LUMO). We then conducted molecular docking and dynamics against TNBC to ascertain binding affinity and stability. Our findings indicated that Scutellarein derivatives, specifically DM03 with a binding energy of -10.7 kcal/mol and DM04 with -11.0 kcal/mol, exhibited the maximum binding tendency against Human CK2 alpha kinase (PDB ID 7L1X). Molecular

**Funding:** The authors received no specific funding for this work.

**Competing interests:** The authors have declared that no competing interests exist.

dynamic simulations were performed for 100 ns, and stability was assessed using root-mean-square deviation (RMSD) and root-mean-square fluctuation (RMSF) parameters, suggesting significant stability for our chosen compounds. Furthermore, these molecules met the pharmacokinetics requirements for potential therapeutic candidates, displaying non-carcinogenicity, minimal aquatic and non-aquatic toxicity, and greater aqueous solubility. Collectively, our computational data suggest that Scutellarein derivatives may serve as potential therapeutic agents for TNBC. However, further experimental investigations are needed to validate these findings.

## 1. Introduction

Breast cancer is a difficult challenge for the global public health community and the disease's growing prevalence [1]. It seems to substantially affect the lives of a large number of females globally by breast cancer [2, 3]. Among different types of breast cancer, triple-negative breast cancer (TNBC), one of the breast cancer subtypes with negative expression of progesterone receptor (PR), estrogen receptor (ER), and human epidermal growth factor receptor-2 (HER2), spread quickly to other body regions which has a higher chance of early decline and death than other subtypes of cancer [4].

Approximately one million women are diagnosed with breast cancer annually, and TNBC accounts for approximately 15–20 percent [5, 6]. According to epidemiological research findings, TNBC is most often diagnosed in young premenopausal women under the age of 40. These makeup around 15–20% of all breast cancer patients [7]. The survival duration for TNBC is lower than those diagnosed with other breast cancer subtypes. Forty percent of fatalities are during the first five years following identification [8, 9]. TNBC is resistant to endocrine therapy and other molecular therapeutic strategies due to the distinct genetic phenotype that makes it susceptible to these treatments [10]. As a result, chemotherapy is the primary therapeutic option; nevertheless, the effectiveness of traditional chemoradiotherapy is relatively poor [10]. In some countries, bevacizumab has been administered with chemotherapeutic medications to treat TNBC; however, patients did not experience a substantial improvement in their overall survival time due to this treatment [11, 12]. Treatment choices for TNBC are more limited than those for other kinds of invasive breast cancer. This is due to the fact that cancer cells do not possess the estrogen or progesterone receptors or enough of the HER2 protein for hormone treatment or targeted HER2 medicines to be effective against the disease [13–15]. So, even though TNBC has been more invasive in women and has affected numerous women worldwide, there is no presently acknowledged treatment for targeting this TNBC [16, 17]. The chemical compound known as Scutellarin is classified as a flavone, which belongs to the phenolic family, and it has a history of use in innovative medicine containing this compound. It has been demonstrated that Scutellarin may stimulate different types of cancer cells to undergo apoptosis when tested *in vitro*. Additionally, Scutellarin has been shown to have a neuroprotective effect on nerve cells [18]. Scutellarin was also found to have a strong inhibitory effect on a drug-resistant strain of HIV-1 cell-to-cell multiplication when tested in a laboratory [19], as well as vigorous antibacterial and antifungal activity [20]. Table 1 displays the efficiency of Scutellarin on the proliferation of several cancer cell lines.

So, the perspective of this research is to develop and identify a potential drug candidate for treating TNBC by structural modification of Scutellarin. In this case, the most powerful computational method is applied, and performed different types of investigations to validate and establish Scutellarin as a potential medication against TNBC. The main advantage of

**Table 1. Effects of Scutellarin on the proliferation of several cancer cell lines and the underlying molecular pathways [21].**

| Cancer cell line | Molecular mechanism of scutellarin | Reference |
|---|---|---|
| Colorectal cancer | Activation of caspase-6 resulted in the stimulation of cancer cell death and cancer cell spread. | [22] |
| Prostate cancer | Induced cell death (apoptosis) and induced cell cycle arrest in the G2/M phase to restrict cell growth. | [23] |
| Breast cancer | Up-regulating the Hippo/YAP signaling pathway led to a suppression of cell proliferation and invasion of the target cell. | [24] |
| Lung cancer | Elevated levels of apoptosis and autophagy in lung cancer cells are caused by activation of the ERK/p53 and c-met/AKT signal transduction pathways | [25] |
| Renal cell carcinoma | Suppressed renal cell carcinoma cell growth and invasion by stimulating phosphatase and tensin homologue expression. | [26] |

computational investigation, it may reduce time, cost, and resources. These techniques may also lower the chances of failure during clinical or pre-clinical trials [27].

Scutellarin is the glycosyloxyflavone that has a significant role as an antineoplastic agent. It binds with PR and ER and impedes the development of cancer cells, which the body finally eliminates. In Fig 1, TNBC cell development occurs and gradually spreads the risk factor. Still, when Scutellarin binds with the estrogen receptor, abnormal cell growth stops, and healthy cell production starts. It is a theoretical and probable mechanism but not established.

## 2. Materials and methodology

### 2.1 Ligand preparation and optimization

For the preparation of ligands, all the chemical structures of Scutellarein derivatives have been designed by ChemBioDraw 12.0 [28]. Then, these Scutellarein derivatives were uploaded in material studio and optimize through the implementation of the DFT (density functional theory) from the DMol3 code [29–31]. The B3LYP functional and DNP basis sets were implemented in DMol3 code to get such exact outcomes [32]. These analytical tools were implemented to establish the frontier molecular orbitals (HOMO, LUMO) and the amplitude of the HOMO and LUMO following optimization. The optimized molecules were exported as PDB file types in order to be used for subsequent computer work, such as molecular docking, molecular dynamic, and ADMET. The magnitude of chemical reactivity and characteristics are approximated by applying relevant and approved algorithms, and the results are listed in **Table 4.** the energy gap; hardness; softness was calculated for the reported drug candidate.

### 2.2 Lipinski rule and pharmacokinetics

Swiss ADME online database has been implemented to evaluate the specified compounds' Lipinski Rule and pharmacokinetic before the molecular docking experiment (http://www.swissadme.ch) [33]. Lipinski rule and pharmacokinetic may be characterized as a complicated balance of several chemical characteristics and structural factors that indicate whether a newly developed molecule is close to existing medications. These features are marked by hydrophobicity, drug likeness, hydrogen bonding characteristics, molecule size and molecular weight, bioavailability, and many more.

### 2.3 ADMET profile prediction

There is a higher prevalence of failure in developing new drugs due to inadequate pharmacokinetic and safety features. Computational techniques may be able to assist in reducing these

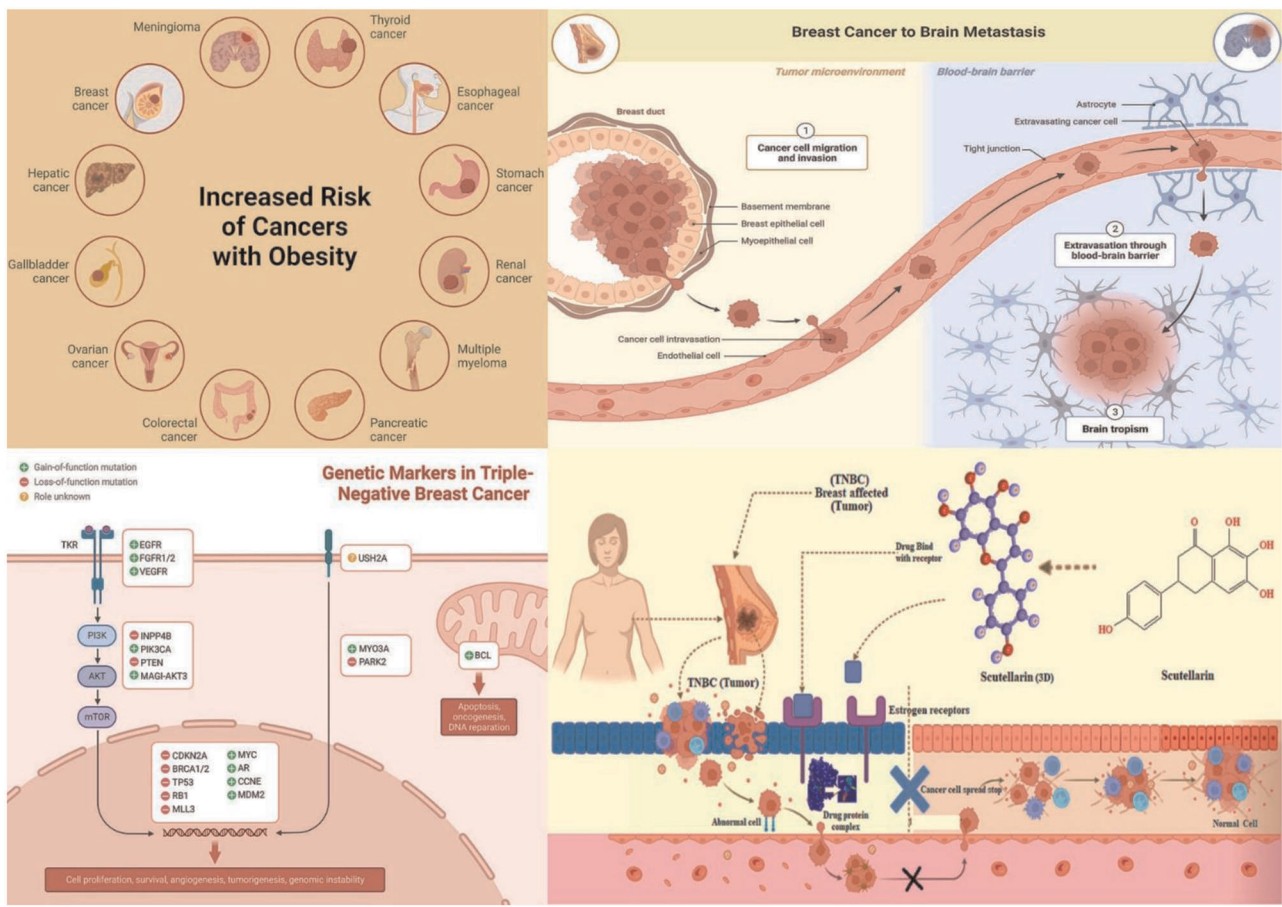

**Fig 1. Risk factors, genetic markers, and probable mechanism of TNBC (Created with Biorender.com).**

concerns. Regarding pharmacokinetic characteristics, pkCSM could be an alternative approach to predicting ADMET (absorption, distribution, metabolism, excretion, and toxicity) features. This is the most preferable resource for estimating ADMET parameters. The report and research documented that ADMET is beneficial for predicting the pharmacokinetics of biomolecules before going to clinical or pre-clinical trials [34–36]. So, using this online system pkCSM "https://biosig.lab.uq.edu.au/pkcsm/" online tool, the ADMET feature was evaluated and analyzed [34].

## 2.4 Method for molecular docking

For docking analysis, the initial three-dimensional (3D) structural triple-negative breast cancer responsible protein (PDB ID: 7L1X & PDB ID 5HA9) was collected from the protein data bank [37, 38]. After that, the Pymol program version V2.3 (https://pymol.org/2/) was implemented to purify the proteins such as water molecules, and unexpected ligands or heteroatoms were removed and obtained a fresh protein. To bind a drug in a specific side, like as lock and key model, must be cleaned or fresh target receptor, excess molecules such as water and other unwanted substances may interfere to bind specific site. So, the water and other unwanted substances are cleaned before docking. Then, they were saved as PDB files and the energy of the chosen receptor was minimized using swisspdbviwer application [39, 40]. After that, the

| Human CK2 alpha kinase (PDB ID 7L1X) | TNBC (PDB ID 5HA9) |
|---|---|
| Method: X-RAY DIFFRACTION | Method: X-RAY DIFFRACTION |
| Ref. [39] | Ref. [40] |

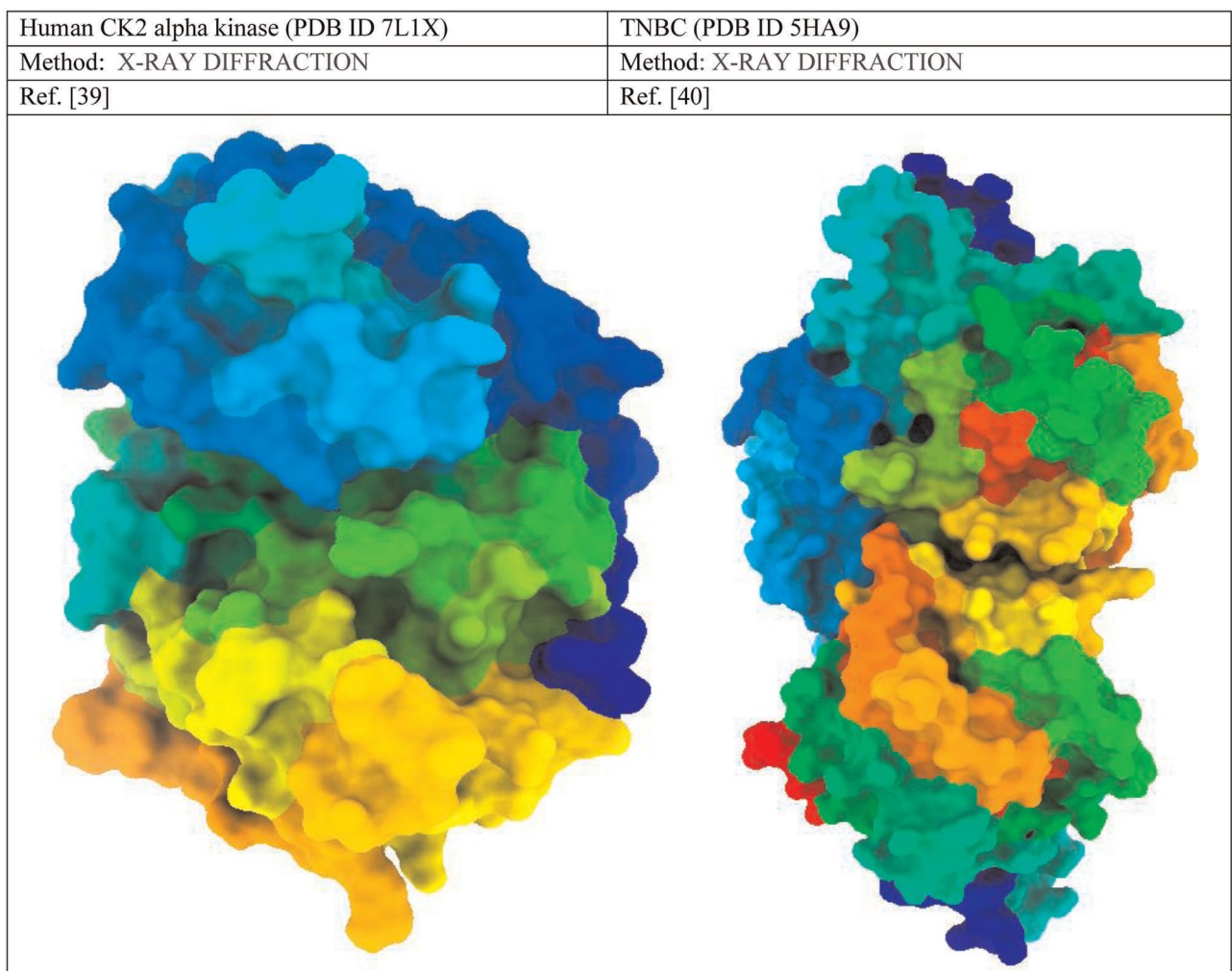

**Fig 2. Three-dimensional protein structure of TNBC and its basic features.**

proteins were imported into the PyRx software and converted into macromolecules; similarly, all the ligands were imported and converted in pdbqt format. Finally, molecular docking has been performed through AutoDock Vina [41]. Then, the dock file was imported into the Pymol application to generate a complex structure. The docked complexes were transferred into BIOVIA Discovery Studio Visualizer for further analysis and visualization of the active amino acid residues and proteins ligands pocket diagram [42]. The three-dimensional structure of the targeted protein is displayed in Fig 2.

### 2.5 Molecular dynamic simulations (MDs)

MD simulations' purpose is to determine chemical compounds' stability during the formation of drug protein complex. So, the highly configured desktop computers equipped with NAMD software were implemented and performed in batch mode or real-time with a live view to run the MD simulation up to 100 ns [43–45]. The AMBER14 force field was implemented in the holo-form (drug-protein) of the MD simulation to provide the optimum fitting or binding pose and stabilization of the ligand-protein docking [46, 47]. The complete system was

adjusted using 0.9 percent NaCl at 298 K temperature and adding a water solvent. During the simulation, a cubic cell was transmitted within 20 Å on either side of the system and the corresponding boundary conditions. After simulation, the root means square deviation (RMSD) and root mean square fluctuation (RMSF) were analyzed using the visual molecular dynamics (VMD) software [48].

## 3. Result and discussions

### 3.1 Designing derivatives of Scutellarin

The primary compound was Scutellarin which has previously been documented as anti-cancer, antimicrobial, and many more pharmacological effects [49, 50]. So, based on the previous investigation and literature review, the Scutellarein structure has been modified to predict the potential anti-cancer efficiency against targeted TNBC. Since the structure-activity relationship is a well-known approach for developing and designing new and innovative molecules to assist in the discovery or synthesis of a novel drug to get desired qualities and activities. The different position of the functional group of Scutellarein has been substituted by the Benzene ring, $OCH_3$, & $NH-CH_2-CH_2-OH$ to obtain better efficacy and potential drug for inhibiting TNBC. The main objective is to identify how different functional groups impact pharmacological activity on Scutellarein against TNBC. The modified chemical structures of TNBC are shown in Fig 3.

### 3.2 Optimized structure of selected Scutellarin derivatives

The material studio application has been implemented to visualize the geometry optimization of the ten bioactive Scutellarein and its modification's structure. The optimized chemical structures of these derivatives are displayed in (Fig 4). The optimized structure represents the symmetry and asymmetry point through the chemical structure and configuration. The goal of lead optimization is to maximize the efficiency of the most promising compounds to maintain the desired properties at optimum conditions [51].

### 3.3 Pass prediction spectrum

Pass is online website has been used to explore pass prediction score(http://way2drug.com/PassOnline/predict.php) [52]. With the studied compounds, this implementation was carried out to detect possible therapeutic qualities that might be validated by laboratory investigation. PASS Prediction value evaluates the potentiality of a novel molecule to the structures of well-known physiologically active substances, allowing researchers to determine whether or not a chemical will have a particular activity. This method can be applied from the very beginning of the investigation or development of new drugs. The previously generated geometries of the described molecules were uploaded to the PASS online tool as a mol form, and the potential mode of action and bioactivities were projected. The probabilities of activity (Pa) and inactivity (Pi) are the basis of these filters. Scientists could conserve time and money by swiftly sorting through thousands of drug candidates utilizing pass prediction algorithms and filters, then focusing on the most promising ones. This has the potential to drastically shorten the time it takes to find new drugs and enhance the likelihood that they will be successful in treating a wide variety of conditions [52, 53].

Regarding data, the probability of being active (Pa) value range is 0.161 to 0.383 for antiviral, 0.182 to 0.402 for antibacterial, 0.236 to 0.515 for antifungal, and the last one is antineoplastic, which is 0.572 to 0.794 (Showing in Table 2). It has been seen that the probability of being active (Pa) values are much more significant for antineoplastic than antiviral,

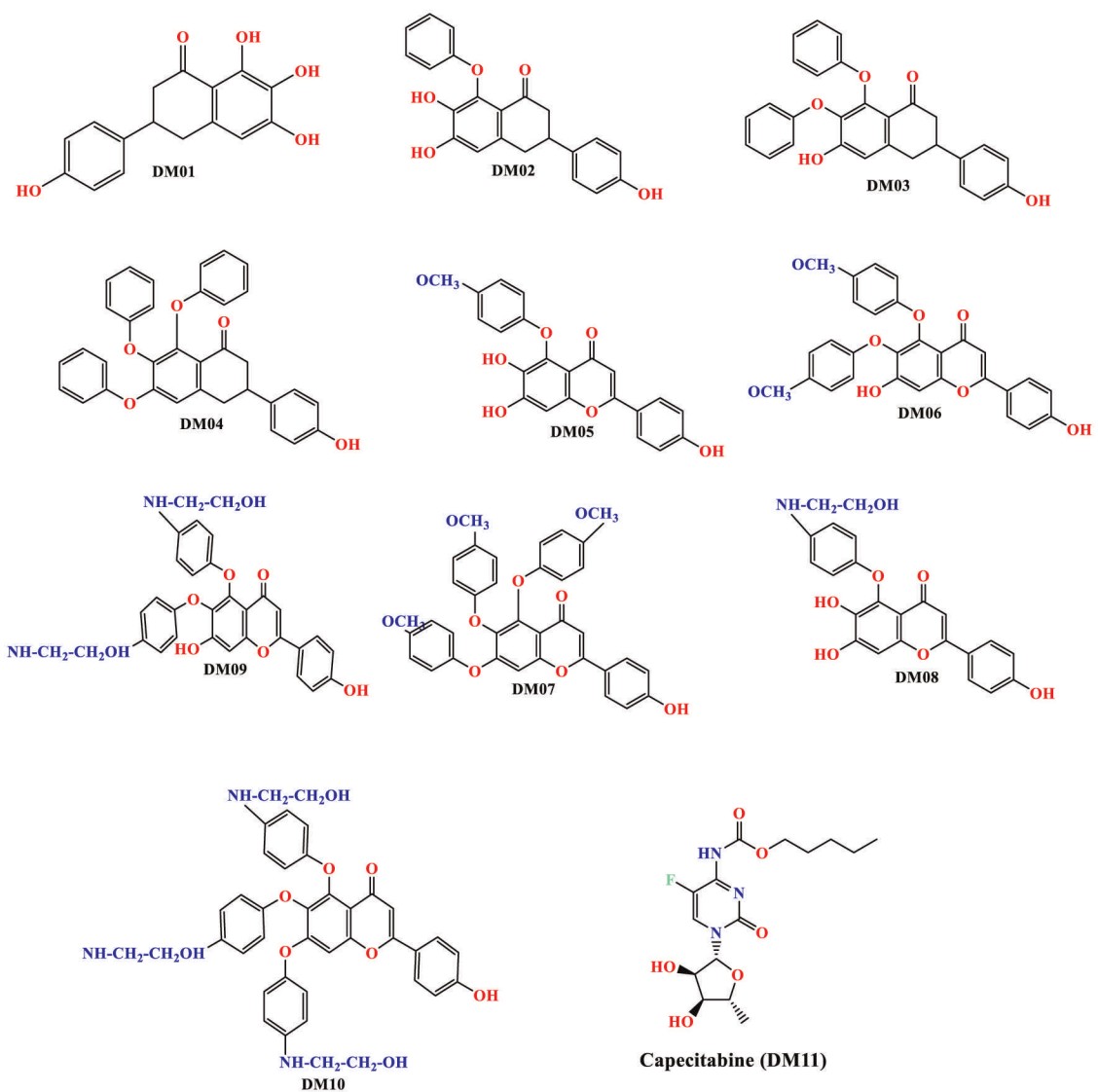

**Fig 3. Modified structure of Scutellarein derivatives.**

antibacterial, and antifungal. So, after observing the data on the probability of being active (Pa), It has been decided to study them against cancer further. So, two proteins of TNBC proteins have been picked up as targeted receptors for further computational analysis.

### 3.4 Lipinski rule analysis for oral medication

Orally active drugs should be relatively small according to the Lipinski rule [54]. Lipinski's rule establishes whether a biochemical molecule with a specified pharmacological or biological activity has molecular qualities and physical features, constituting it a probable orally administered medication in mammals [54].

According to this concept, medicine is considered to have suitable oral medication if they should fulfill this criterion of Lipinski rule. So, in our predicted data, it has been seen that the molecular weight of the compounds is below 286.24 to 691.73, and the range of

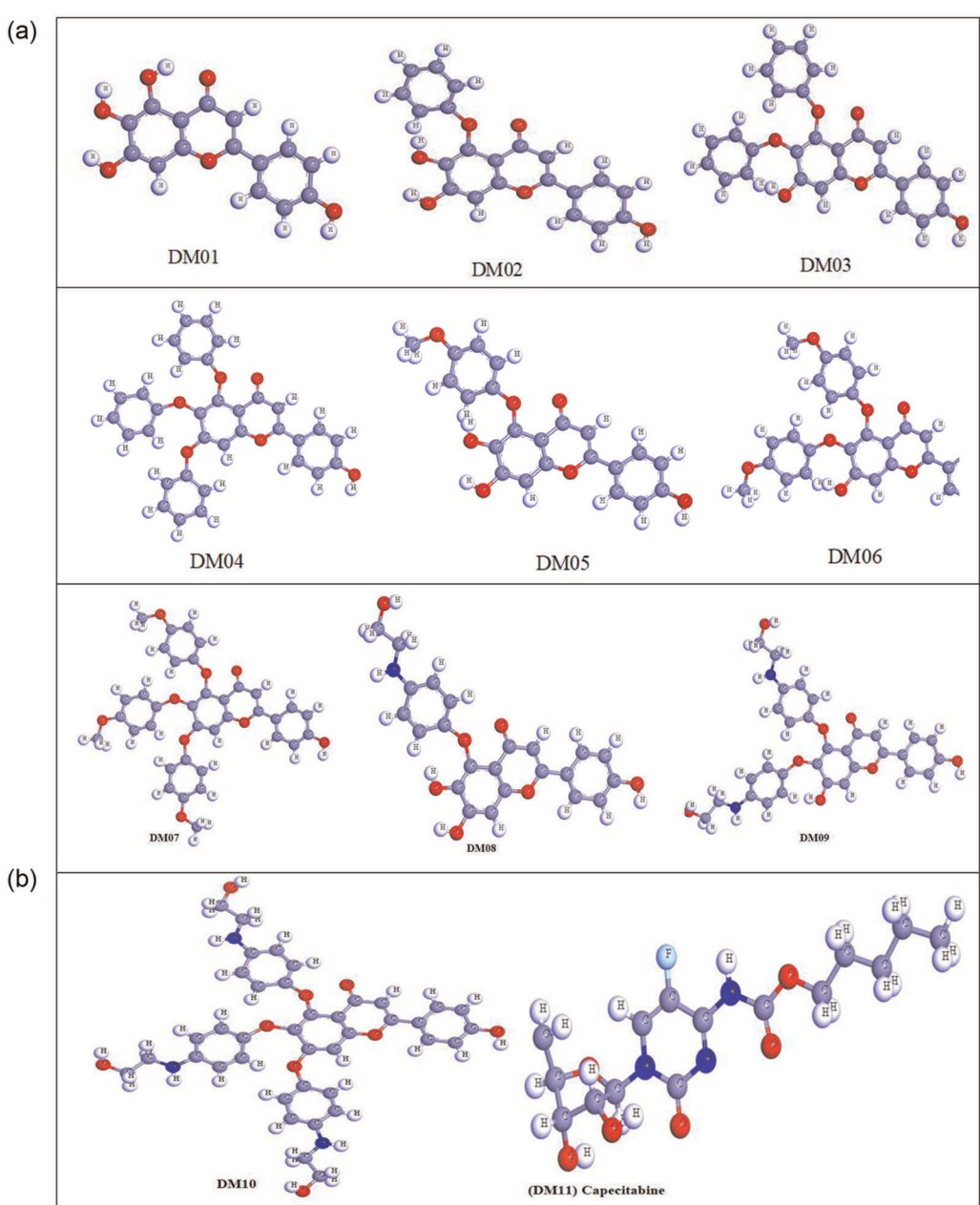

**Fig 4. Three dimensional structure of Scutellarein derivatives.**

topological polar surface area is 78.13–174.91. Besides, the bioavailability scores are better for all compounds (0.55); only ligands DM09 & DM10 are poor bioavailability scores. Another essential parameter is the GI absorption rate, which indicates that most drugs are poorly absorbed in the GI tract but the molecules DM01, DM02 & DM05 have high GI absorption rates (Table 3). Finally, it is observed that all drugs have satisfied the Lipinski rule where only two molecules declined by Lipinski rule DM09 & DM10 due to high molecular weight. So, ignoring the molecular weight, they should be recommended as potential oral medication.

**Table 2. Computational data of pass prediction spectrum.**

| Ligand No | Antiviral | | Antibacterial | | Antifungal | | Antineoplastic | |
|---|---|---|---|---|---|---|---|---|
| | Pa | Pi | Pa | Pi | Pa | Pi | Pa | Pi |
| DM01 | 0.314 | 0.032 | 0.402 | 0.029 | 0.515 | 0.028 | 0.794 | 0.013 |
| DM02 | 0.204 | 0.092 | 0.342 | 0.045 | 0.496 | 0.031 | 0.687 | 0.028 |
| DM03 | 0.162 | 0.145 | 0.278 | 0.069 | 0.442 | 0.041 | 0.665 | 0.032 |
| DM04 | 0.381 | 0.044 | 0.264 | 0.076 | 0.405 | 0.049 | 0.587 | 0.087 |
| DM05 | 0.161 | 0.147 | 0.326 | 0.051 | 0.487 | 0.033 | 0.698 | 0.026 |
| DM06 | 0.383 | 0.043 | 0.260 | 0.078 | 0.431 | 0.043 | 0.679 | 0.030 |
| DM07 | 0.355 | 0.057 | 0.245 | 0.086 | 0.392 | 0.051 | 0.615 | 0.042 |
| DM08 | 0.161 | 0.147 | 0.252 | 0.082 | 0.319 | 0.074 | 0.644 | 0.036 |
| DM09 | 0.356 | 0.056 | 0.193 | 0.124 | 0.267 | 0.097 | 0.628 | 0.039 |
| DM10 | 0.325 | 0.075 | 0.182 | 0.135 | 0.236 | 0.114 | 0.572 | 0.051 |

## 3.5 Molecular orbitals and chemical reactivity descriptor

Chemical reactivity and kinetic stability are implied to be represented by frontier orbitals of a molecule, and it is the essential orbitals feature in molecules bioactivity. There are two types of frontier orbitals in molecules: HOMO and LUMO. The shift of the electron from the lowest to the highest energy state is mainly accounted for by the excitement of one electron from HOMO to LUMO [55]. Consequently, the excitement of electrons from the HOMO to the excited state LUMO implies significantly more energy. The kinetic stability of a molecule is a linear relationship between the HOMO-LUMO energy gap, which is described as increasing the HOMO-LUMO energy gap, simultaneously growing the chemical reactivity and kinetic stability [56].

Table 4 displays the computed values of molecular orbital energies, including the two well-recognized chemical variables of the energy gap, Chemical potential, Electronegativity, Hardness, and Softness. The documented compound DM01 & DM02 implied the highest HOMO--LUMO energy gap (7.003 eV), which regards them as a more stable configuration. Besides, the derivatives with the maximum softness value were estimated to be 1.1608 in Ligand DM04, which means this drug could be dissolved more quickly Table 4. Simultaneously, the

**Table 3. Predicted data of Lipinski rule.**

| Ligand No | Number of rotatable bonds | Hydrogen bond acceptor | Hydrogen bond donor | Topological polar surface area, Å² | Consensus Log Po/w | Lipinski rule | | Molecular weight | Bioavailability Score | Gastrointestinal absorption |
|---|---|---|---|---|---|---|---|---|---|---|
| | | | | | | Result | violation | | | |
| DM01 | 01 | 06 | 04 | 111.13 | 1.81 | Yes | 00 | 286.24 | 0.55 | High |
| DM02 | 03 | 06 | 03 | 100.13 | 3.16 | Yes | 00 | 362.33 | 0.55 | High |
| DM03 | 05 | 06 | 02 | 89.13 | 4.68 | Yes | 00 | 438.43 | 0.55 | Low |
| DM04 | 07 | 06 | 02 | 78.13 | 6.20 | Yes | 01 | 514.52 | 0.55 | Low |
| DM05 | 04 | 07 | 03 | 109.36 | 3.17 | Yes | 00 | 392.36 | 0.55 | High |
| DM06 | 07 | 08 | 02 | 107.59 | 4.67 | Yes | 00 | 498.48 | 0.55 | Low |
| DM07 | 10 | 09 | 01 | 105.82 | 6.20 | Yes | 01 | 604.60 | 0.55 | Low |
| DM08 | 07 | 07 | 05 | 132.39 | 2.57 | Yes | 00 | 421.40 | 0.55 | Low |
| DM09 | 11 | 08 | 06 | 153.65 | 3.55 | No | 02 | 556.56 | 0.17 | Low |
| DM10 | 16 | 09 | 07 | 174.91 | 4.46 | No | 03 | 691.73 | 0.17 | Low |

**Table 4. Data of chemical descriptors calculation.**

| S/N | A = -LUMO | I = - HOMO | Energy Gap E(gap) = I-A | Hardness | Softness |
|---|---|---|---|---|---|
| DM01 | -1.645 | -8.648 | 7.003 | 3.5015 | 0.2856 |
| DM02 | -1.561 | -8.564 | 7.003 | 3.5015 | 0.2856 |
| DM03 | -1.579 | -8.205 | 6.626 | 3.313 | 0.3018 |
| DM04 | -3.569 | -5.292 | 1.723 | 0.8615 | 1.1608 |
| DM05 | -3.422 | -6.651 | 3.209 | 1.6045 | 0.6231 |
| DM06 | -1.748 | -8.445 | 6.697 | 3.3485 | 0.2986 |
| DM07 | -1.674 | -8.426 | 6.752 | 3.3760 | 0.2962 |
| DM08 | -1.572 | -7.494 | 5.922 | 2.961 | 0.3377 |
| DM09 | -1.568 | -7.576 | 6.008 | 3.004 | 0.3329 |
| DM10 | -1.547 | -7.543 | 5.996 | 2.998 | 0.3336 |

maximum hardness is about 3.5015 in Ligand DM01 & DM02, and this hardness data indicates that they might have required more time to break after reaching the physiological system. In addition, the stability of drugs, such as reactive species, molecules, and their propensity to molecularly react to create new substances, incorporate new physiochemical phenomena, and so on are all determined by their chemical potential under the most prevalent thermodynamic circumstance of constant temperature and pressure [57].

### 3.6 Frontier molecular orbitals (HOMO and LUMO)

The HOMO and LUMO orbitals have a crucial role in determining the efficiency of kinetics and having a biological impact on protein interactions [58]. The HOMO and LUMO orbital geometries describe an electron-rich area and electron deficiency area. They were determined using the DFT method. Fig 5 shows that LUMO is often encountered in cation regions, notably in the positive atom-containing compartments where the positive charge has been located. Thus, an additional electron is attached to the LUMO segment. On the other hand, HOMO is often encountered in anion segments, notably in the negative atom-containing compartments where the electronegative charge could be presented.

To illustrate, the yellow and deep maroon moiety displayed the LUMO or positive charge region, and the bright greenish shade revealed the HOMO segment or negative charge portion. It is often thought that the protein may be coupled to the LUMO portion of the molecules [59, 60].

### 3.7 Molecular of Electrostatic Potential (MEP) charge distribution mapping

The MEP has gained a compatibility aspect that shows the most promising zones for the targeted electrophile and nucleophile of charged area molecules on organic materials [61]. The MEP clarifies the biological mechanism and the H-bonding coupling or interaction [62] and it is significant in the investigation of describing organic molecules. The electrostatic potential map represents a simple approach for determining the interactions between different molecular geometries. In this study, the electrostatic potential map of the targeted compound has been simulated using B3LYP, with a basis set of 3-21G, and optimized the structure as an outcome (Fig 6). It displays the molecular structure and size and positive, negative, and neutral electrostatic probability regions by displaying the color difference. Besides, it is also a

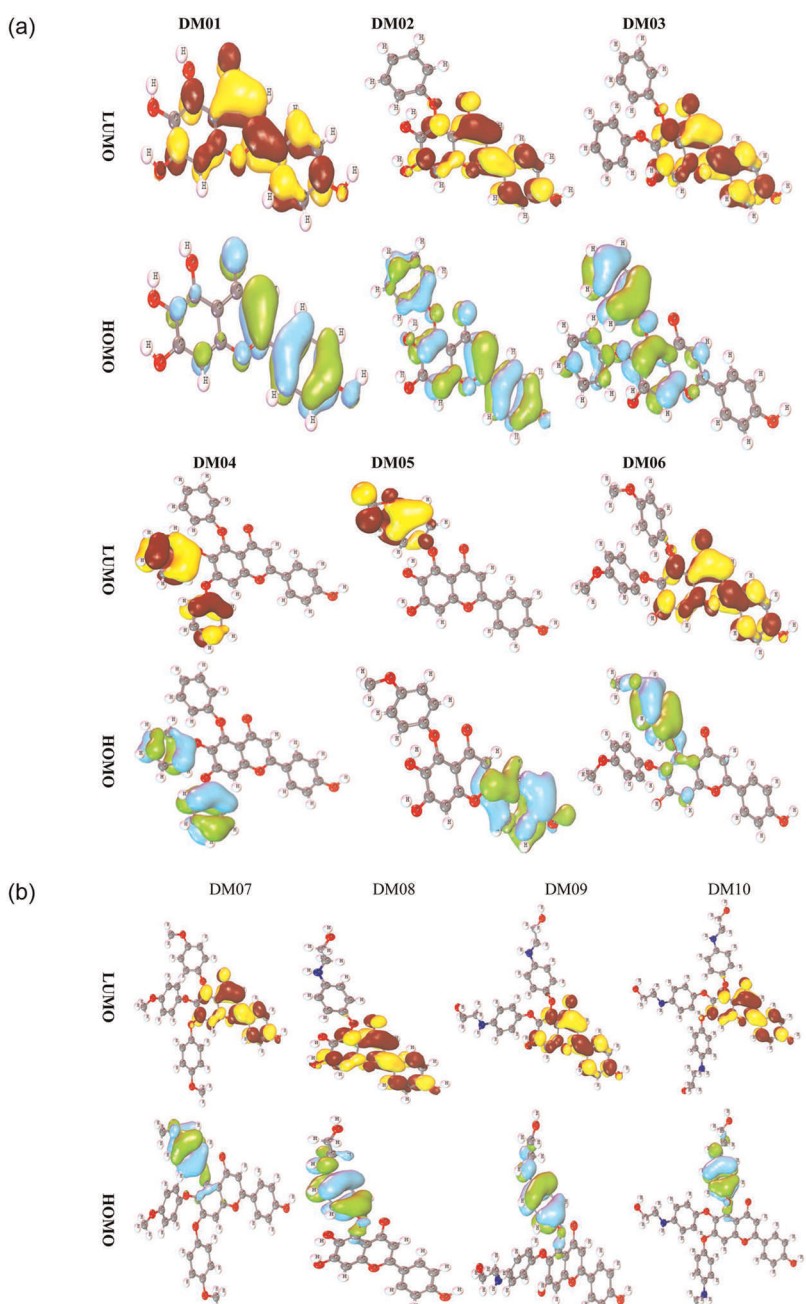

**Fig 5. Frontier molecular orbitals (HOMO and LUMO) diagram.**

prominent approach to investigating the relationship between physicochemical characteristics and the structure of the targeted compound [63, 64].

The assaulting region's potential declines in the following manner: blue, red, and white. The white region is indicated as the neutral region where no assault occurs. The red color shows the high electron saturation area, and this zone described that electrophile might readily assault. The minimal electron density surface is represented by the blue hue, which is amenable to nucleophilic assault.

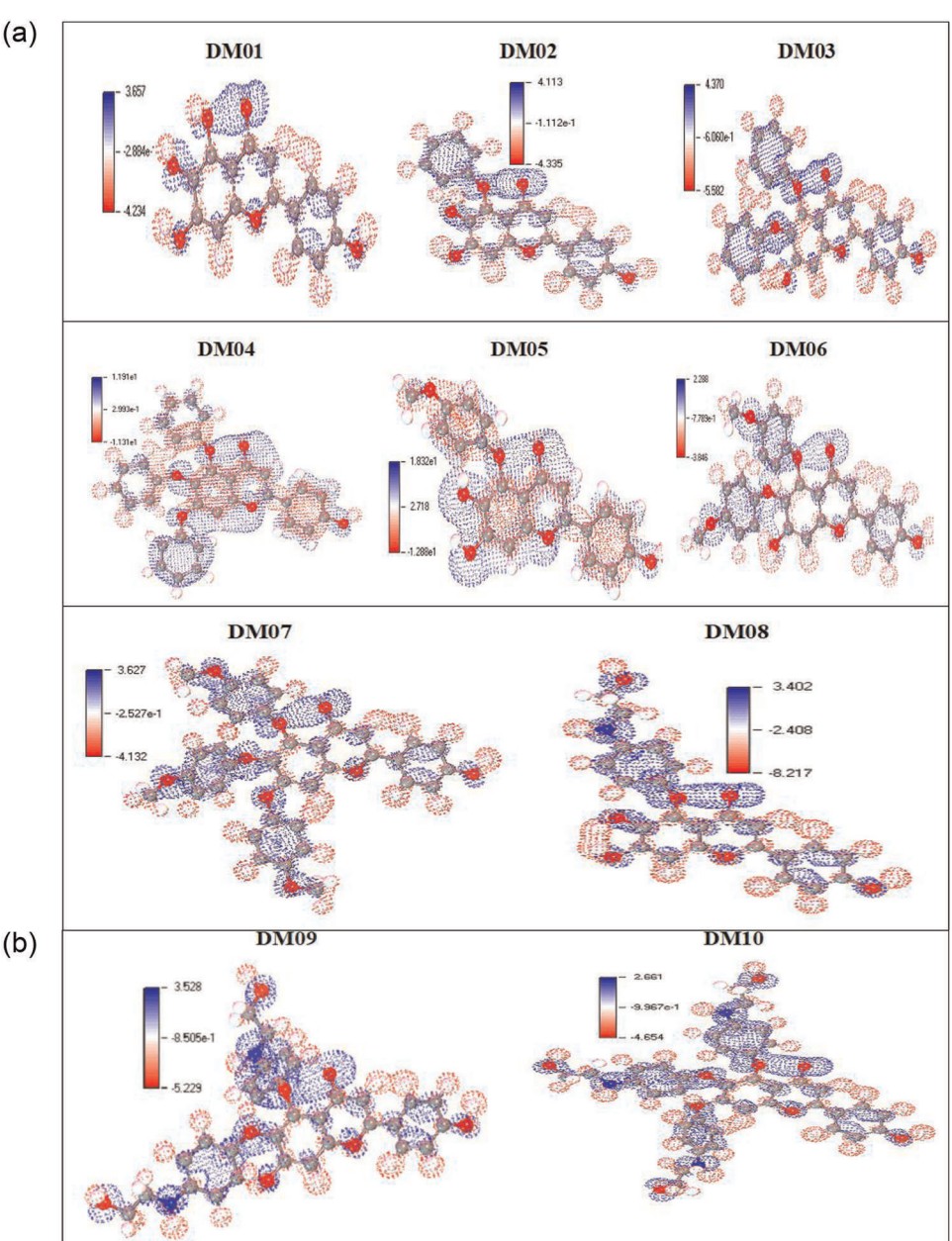

**Fig 6. MEP mappings.**

## 3.8 Molecular docking against targeted receptor

The molecular docking technique has utilized to mimic the interaction between a small molecule and a protein at the atomic level, providing insight into the underlying efficiency involved and allowing scientists to define the binding energy of small molecules in the binding region of target proteins [65–67]. The docking computations were conducted with the help of the program AutoDock (PyRx), and the binding energy of the peptide was determined. The standard binding affinity has been considered for an active drug molecule -6.0 kcal/mol or more [68]. The Scutellarein derivatives (DM03, DM04, DM06, and DM07) were found to have the

**Table 5.  Data of binding energy against TNBC.**

| Drug Molecules No | Human CK2 alpha kinase (PDB ID 7L1X) | | | TNBC receptor (PDB ID 5HA9) | | |
|---|---|---|---|---|---|---|
| | Binding Affinity (kcal/mol) | No of H Bond | No of Hydrophobic Bond | Binding Affinity (kcal/mol) | No of H Bond | No of Hydrophobic Bond |
| DM01 | -9.5 | 04 | 11 | -9.4 | 02 | 05 |
| DM02 | -9.7 | 01 | 11 | -10.5 | 04 | 04 |
| DM03 | -10.7 | 01 | 11 | -8.2 | 03 | 06 |
| DM04 | -11.0 | 02 | 07 | -7.6 | 02 | 06 |
| DM05 | -9.6 | 04 | 11 | -10.5 | 02 | 06 |
| DM06 | -10.0 | 02 | 12 | -7.7 | 02 | 09 |
| DM07 | -10.0 | 04 | 17 | -8.1 | 02 | 09 |
| DM08 | -9.2 | 03 | 12 | -10.0 | 06 | 04 |
| DM09 | -9.1 | 06 | 11 | -7.5 | 05 | 07 |
| DM10 | -8.8 | 06 | 12 | -7.6 | 08 | 04 |
| Standard Capecitabine | -7.7 | 03 | 03 | -7.5 | 02 | 06 |

maximum bioactivity against Human CK2 alpha kinase (PDB ID 7L1X) and compounds (DM02. DM05 & DM08) have shown the most potency compared to the all-other derivatives (Table 5).

For Human CK2 alpha kinase (PDB ID 7L1X), the most significant binding affinity has been seen at -10.7 kcal/mol and -11.0 kcal/mol in ligand DM03 and DM04, and the most considerable binding affinity has been seen at -10.5 against (PDB ID 5HA9) in ligand DM02 and ligand DM05. So, to compare with standard drugs, capecitabine has also been studied with the designing ligands. It is found that the new drug molecules are more active than the FDA-approved drugs capecitabine. Now, it can be said that the drug molecules are opposed to the standard binding affinity and actively bind with triple-negative breast cancer protein as a potential inhibitor. So, further experimental studies might be conducted to validate this theoretical investigation.

### 3.9 Protein-ligand interaction

The interaction diagrams of the drug-protein combinations, hydrogen bonding, and molecular docking pocket have been developed utilizing the BIOVIA Discovery Studio and Pymol application software. The hydrogen bond interactions such as conventional and non-conventional H bonds, hydrophobic interactions including pi-sigma, alkyl, and pi-alkyl interactions, hydrogen bond donor and hydrogen bond -acceptor interactions have been generally looked at protein and ligand interaction and hydrogen bond and hydrophobic bonds are played significant rule for drug activity. Different engagement and binding energies between the substance and its desired targeted protein are represented graphically in Fig 7.

In hydrogen bonding, the red color is described acceptor region whereases the acceptor region describes the sky-blue color. Besides, the 2d image of active amino acid residues is seen that A: TYR-50, A: HIS-160, A: ASP-175, A: LEU-45, A: GLY-48, A: VAL-66, A: VAL-53, A: ILE-95, and A: ILE-174 is generated for triple negative Breast cancer (PDB ID 7L1X) with DM04, whereases A: TYR-50, A: HIS-160, A: ASP-17,5 A: LEU-45, A: GLY-48 A: VAL-66, A: VAL-53, A: ILE95, and A: ILE-174 is generated during the formation of the complex with PDB ID 5HA9. Besides, the amino acid residues and their bond distance for all compounds are given in the S1 Table.

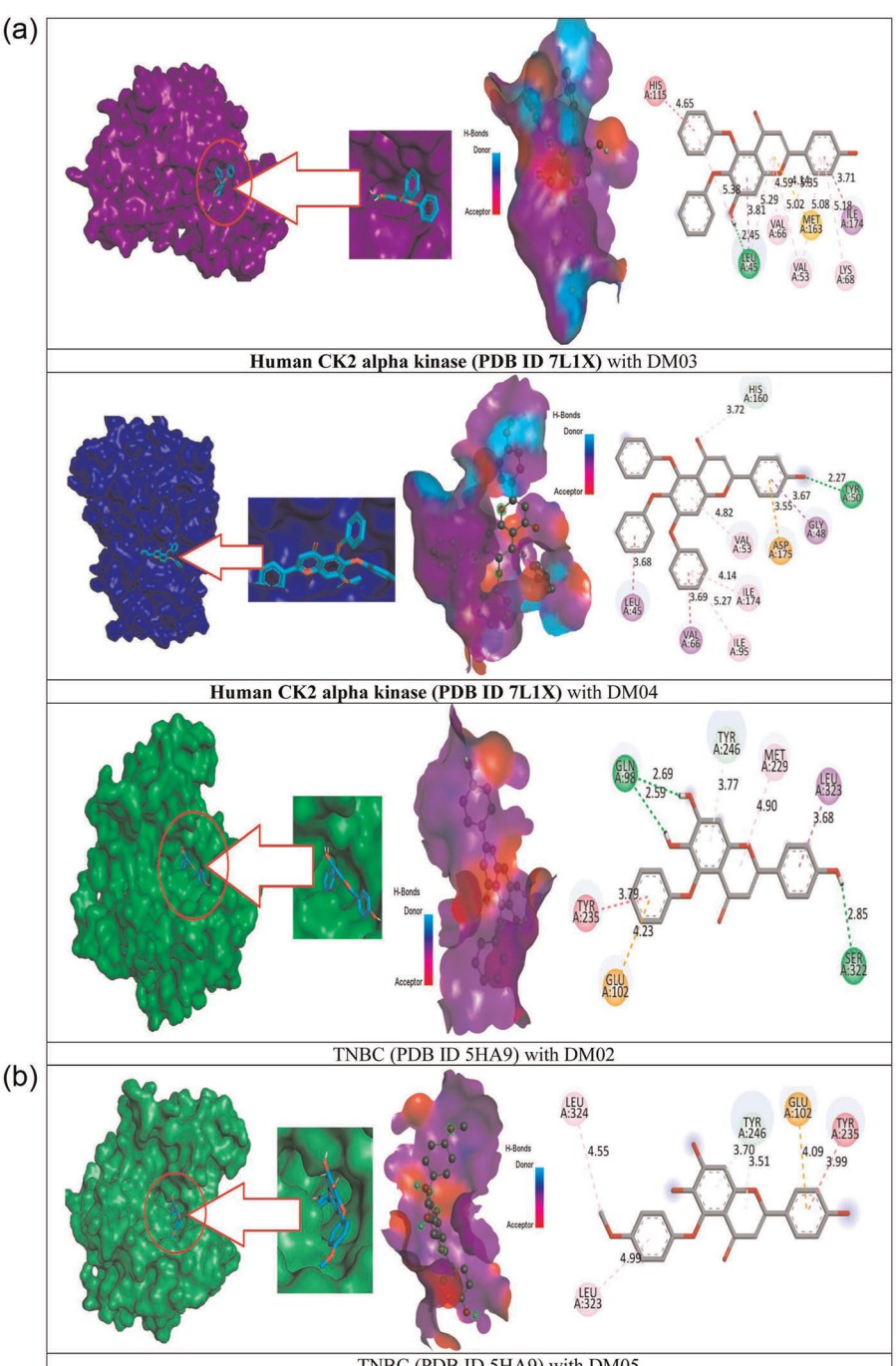

**Fig 7. The results of molecular docking pose interactions between compounds and proteins.**

## 3.10 Molecular dynamic simulation

Molecular dynamics have been simulated protein-ligand complexes to express high stabilization at the simulation point of the compounds and determine the accuracy docking procedure in terms of the average RMSD and RMSF. This value represents the binding pose in the respective crystal structures, ligand, and protein interaction complex. The RMSD and RMSF drift

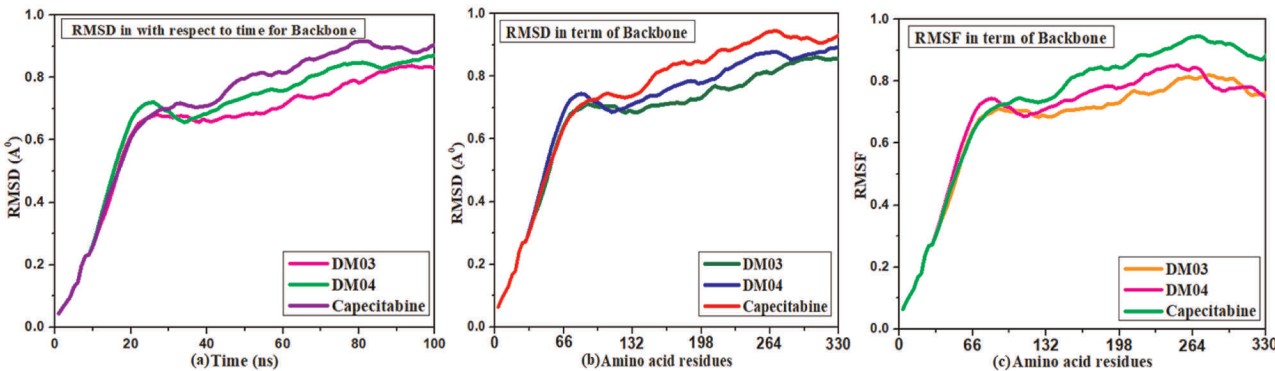

**Fig 8. MD simulation and RMSD, RMSF results of TNBC with ligand.**

assessment statements have been implemented to determine the molecule's stability. Fig 8 illustrates the diagram. The MD simulation point indicates that the drug complex has excellent engagement with targeted protein.

For TNBC (PDB ID 7L1X), the RMSD scatterplot in Fig 8(a) demonstrates that the dynamic evaluation of picked molecules from the beginning to the end position is consistent from 0.8 Å to 1.0 Å throughout the simulation period. RMSD of docked complexes has been run at 100ns. Again, the RMSD in terms of amino acid backbone Fig 8(b), 0.9 Å, and RMSF is 0.8 Å as maximum Fig 8(c). It has been observed that the range of MD simulation is much lower for the newly designed compounds DM03 & DM04 compared to the standard drug capecitabine. This value indicates the reported drug is highly stable.

## 3.11 Radius of gyration (Rg) and solvent accessible surface area (SASA) analysis

To examine the local conformational stability of the systems, the radius of gyration (Rg) of the backbone atoms of free and bound receptors was computed. In Fig 9, the radius of gyrations as a function of time is depicted. In Fig 9A, upon binding of the DM03, the radius of gyration of the backbone atoms showed a modest increase, indicating a less compact structure after the simulations when compared to the DM04 and capecitabine. Fig 9B shows that after simulations, unlike capecitabine (Fig 9C), the radius of gyration of backbone atoms and the complex did not rise as much upon binding of the DM04. Complexes of proteins and ligands The Rg values of DM04 and the standard medicine capecitabine were greater than those of DM03, which indicates that DM03 had relatively less compacted folding than DM04 and capecitabine.

With the aid of SASA, one can determine the surface area of proteins that a solvent can contact. The data showed that the largest surface area for DM03 occurred in the final 20 ns, while the same time period for DM04 and capecitabine was about 20 and 35 ns, respectively. It is clear from this that DM03 and DM04 acids will bind to the protein complex with a higher degree of efficiency compared to capecitabine (Fig 10).

## 3.12 ADMET profile prediction

ADME-related features such as therapeutic absorption, distribution, metabolism, solubility, and even oral bioavailability are influenced by ADMET characteristics. The computational inspection approaches might be used in the drug development process to predict ADMET features. Water solubility is the most potent criterion for oral drug delivery systems in modern

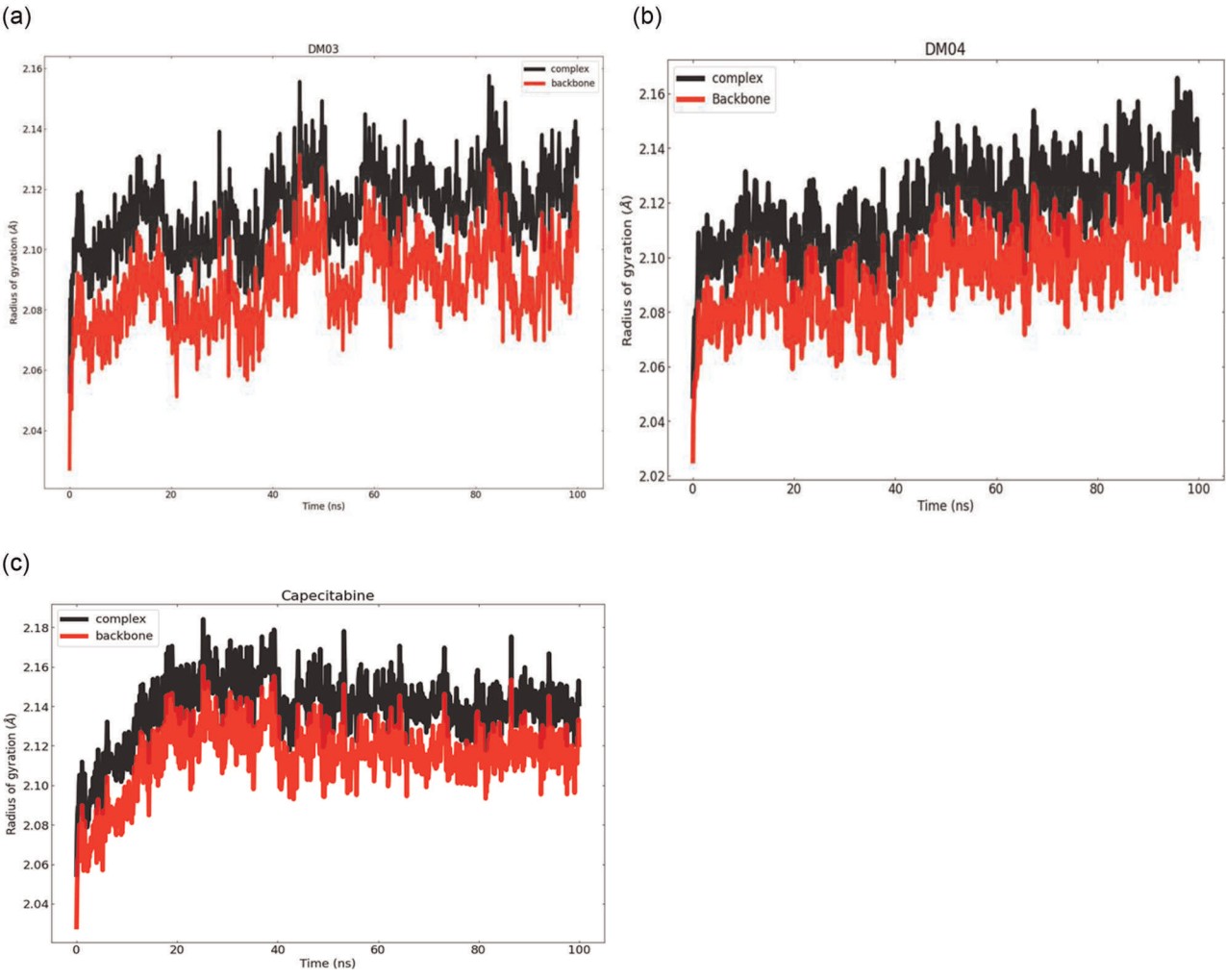

**Fig 9. During 100 ns molecular dynamics simulations for proteins and complexes, the radius of the gyration graph displays versus time plots.**

drug discovery. High water-soluble oral drugs consistently have excellent oral bioavailability with maximum absorption capability [69, 70]. In order to estimate solubility in water for active drug candidates, there is a great interest in developing quick, reliable, structure-based approaches.

In the listed ADMET feature, the water solubility of each compound is a different value. The ligands DM04, DM07, DM08 & DM10, are highly soluble in water, and their range is from -2.893 to -2.998 since the actual values of water solubility (Log S) for slightly soluble vary from -4 to -6, and high solubility compounds -2 to -4, correspondingly [71]. The remaining compounds are little soluble in water which means they are highly soluble in fatty material or lipid. Another absorption parameter is Caco-2 permeability. The standard range of good Caco-2 Permeability is >0.90 according to the pkCSM web tool [72]. In our finding, all drugs have an excellent rate in Caco-2 Permeability, excluding DM04 since it has been higher Caco-2 Permeability than 0.9.

Table 6 lists the distributional features of all substances, including their volume distribution and blood-brain barrier (BBB) permeability. The volume distribution of medicine is closely

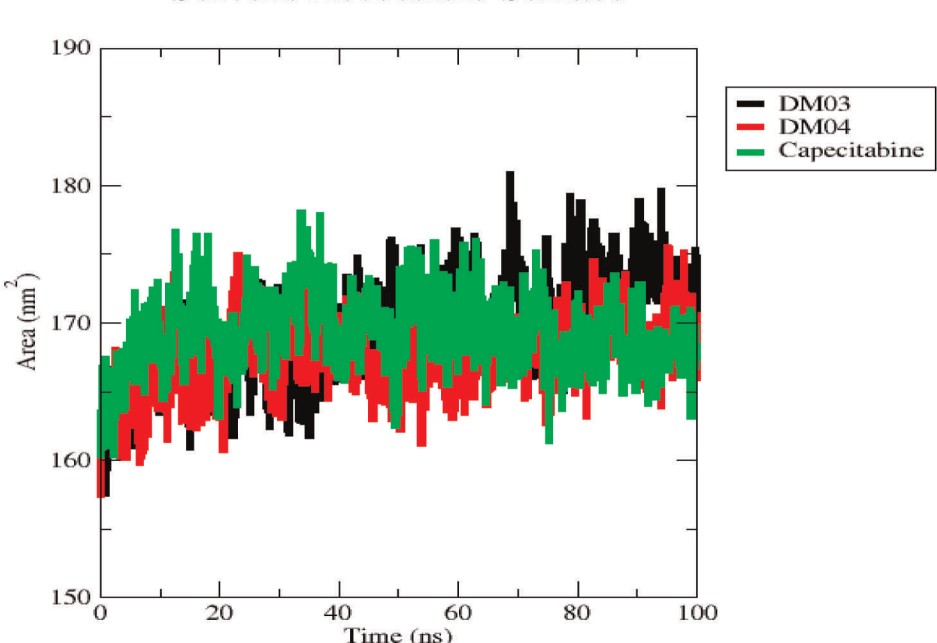

**Fig 10. A plot of solvent accessible surface area (SASA) of the three protein complexes.** DM03 (black), DM04 (red0), and capecitabine (red).

related to medication distribution. The volume of distribution (VD) of a drug throughout blood and tissue should be used to estimate whether the drug distributes uniformly or not uniformly. Lower VD indicates that the drug concentration in plasma is more significant, and the medication cannot diffuse to the tissues as effectively [73]. In this case, most molecules had low VD levels, while some had intermediate VD values. Another parameter, the BBB, is a protective barrier that inhibits unwanted materials from entering the brain and central nervous system (CNS) [74]. In our finding, no drug molecules can permeability BBB.

**Table 6. ADMET profiling prediction of the studied ligands.**

| Molecules No | Absorption | | Distribution | | Metabolism | | Excretion | |
|---|---|---|---|---|---|---|---|---|
| | Water solubility Log S | Caco-2 Permeability $10^{-6}$ cm/s | VDss (human) | BBB permeability | CYP450 1A2 Inhibitor | CYP450 2C9 substrate | Total clearance (ml/min/kg) | Renal OCT2 substrate |
| DM01 | -3.064 | -0.004 | -0.193 | No | Yes | No | 0.511 | No |
| DM02 | -4.235 | 0.416 | -1.10 | No | Yes | Yes | 0.311 | No |
| DM03 | -3.443 | 0.915 | -1.267 | No | No | Yes | 0.385 | No |
| DM04 | -2.977 | 0.990 | -0.87 | No | No | Yes | 0.466 | No |
| DM05 | -3.087 | 0.85 | -0.584 | No | Yes | Yes | 0.718 | No |
| DM06 | -3.101 | 0.224 | -1.307 | No | No | Yes | 0.792 | No |
| DM07 | -2.893 | -0.159 | -0.637 | No | No | Yes | 0.973 | No |
| DM08 | -2.998 | 0.677 | -0.388 | No | Yes | Yes | 0.668 | No |
| DM09 | -3.029 | 0.26 | -1.301 | No | No | Yes | 0.664 | No |
| DM10 | -2.896 | -0.472 | -0.851 | No | No | No | 0.643 | No |
| Capecitabine | -2.939 | 0.339 | -0.253 | No | No | No | 1.138 | No |

On the other hand, all the ligands may actively metabolize in CYP450 2C9 substrate while only DM01, DM02, DM05 & DM08 are inhibited by CYP450 1A2. Finally, the total clearance range is about 0.220 (ml/min/kg) to 0.792 (ml/min/kg) which mean maximum 0.792 (ml/min/kg) drug can out from the body and no drug can excrete through Renal OCT2 substrate. This computational ADMET projection research suggests that the new molecules DM01–DM10 are biologically better drug candidates, which should be considered promising as new drug molecules.

### 3.13 Aquatic and non-aquatic toxicity

Another parameter is aquatic and non-aquatic toxicity. Aquatic and non-aquatic toxicity is a significant objective in assessing undesirable impacts on the environment or human use [75]. In this investigation, a series of computational methods for evaluating biochemical aquatic and non-aquatic toxicity have been discussed with different parameters such as AMES toxicity, oral rat acute toxicity (LD50), oral rat chronic toxicity, etc. In this case, it has been proven that all the drugs are free from AMES toxicity without DM02. The maximum tolerated dose for a human is 0.781 mg/kg/day in DM01. At the same time, the lowest maximum tolerated dose is 0.358 mg/kg/day in Ligand DM09, which indicates that the highest dose of 0.781 mg/kg/day should be taken for a given period (24 Hours); otherwise, the adverse effect may produce. According to current estimates, the acute oral toxicity and the oral rat chronic toxicity of the active compounds vary, with values ranging from 2.418 mg/kg/day to 3.43 mg/kg/day being documented for the oral rat acute toxicity, where the oral rat chronic toxicity being reported from -0.086 mg/kg/day to 4.029 mg/kg/day. Finally, they all are free from skin sensitization with the lowest T. pyriformis toxicity. So, it can be concluded that they might have no adverse effect on aquatic and non-aquatic and could be performed further experimental works in laboratories. Aquatic and non-aquatic poisoning are all of the medications listed in Table 7.

### 4. Conclusion

Our computational analysis aimed to determine the potential therapeutic impact of Scutellarin derivatives against Triple-Negative Breast Cancer (TNBC) and to assess the influence of different functional group additions on binding affinity. The outcomes of this investigation suggest promising efficacy of novel Scutellarin derivatives against TNBC. As highlighted earlier, the Prediction of Activity Spectra for Substances (PASS) results we acquired for our compounds demonstrated a stronger antineoplastic activity than antiviral, antifungal, and antibacterial

**Table 7. Summary of the aquatic and non-aquatic toxicity properties of the studied ligands.**

| S/N | AMES toxicity | Max. tolerated dose (human) mg/kg/day | Oral Rat Acute Toxicity (LD50) (mol/kg | Oral Rat Chronic Toxicity (mg/kg/day) | Skin Sensitization | T. Pyriformis toxicity (log ug/L) |
|---|---|---|---|---|---|---|
| DM01 | No | 0.781 | 2.347 | 2.911 | No | 0.33 |
| DM02 | Yes | 0.523 | 2.699 | 2.225 | No | 0.286 |
| DM03 | No | 0.422 | 3.432 | 0.424 | No | 0.285 |
| DM04 | No | 0.439 | 2.741 | -0.086 | No | 0.285 |
| DM05 | No | 0.566 | 2.418 | 1.740 | No | 0.285 |
| DM06 | No | 0.360 | 2.719 | 1.799 | No | 0.285 |
| DM07 | No | 0.426 | 2.484 | -0,197 | No | 0.285 |
| DM08 | No | 0.671 | 2.443 | 1.758 | No | 0.285 |
| DM09 | No | 0.358 | 2.552 | 2.155 | No | 0.285 |
| DM10 | No | 0.422 | 2.445 | 4.089 | No | 0.285 |
| Capecitabine | No | 1.078 | 2.029 | 2.381 | No | 0.293 |

activities. This led us to focus predominantly on TNBC. For TNBC (PDB ID 7L1X), the docked compounds of modified molecules displayed high binding affinities and remarkable non-bonding interactions (DM03–10.7 kcal/mol, DM04–11.0 kcal/mol, with the most substantial binding affinity observed at -10.5 against (PDB ID 5HA9) for ligand DM02 and ligand DM05). The calculated maximum softness values of the drugs have been reported as 1.1608, suggesting their rapid metabolism upon entering the body. Modified Scutellarin derivatives, such as capecitabine, are projected to have higher reactivity than traditional ligands due to their smaller HOMO-LUMO energy gaps. From an ADMET perspective, the theoretical findings suggested that the new Scutellarin derivatives are safer than reference drugs in terms of toxicity, water solubility, distribution volume, and clearance rate. Importantly, they showed no carcinogenic effects, except for ligand DM02. The improved ADMET properties of Scutellarin derivatives compare favorably with gold-standard drugs, highlighting their potential as a safer treatment alternative. In a nutshell, our computational exploration suggests that Scutellarin derivatives could be valuable and promising therapeutic candidates against TNBC. We recommend further experimental work to explore the potential of Scutellarin for synthetic drug development aimed at combating this lethal form of triple-negative breast cancer.

## Supporting information

**S1 Table.**
(DOCX)

## Acknowledgments

Author would like to acknowledge Department of Pharmacy, Faculty of Allied Health Sciences, Daffodil International University, Daffodil Smart City, Birulia, Savar, Dhaka-1216, Bangladesh.

## Author Contributions

**Conceptualization:** Shopnil Akash.

**Data curation:** Shopnil Akash, Farjana Islam Aovi, Md. A. K. Azad, Ajoy Kumer, Unesco Chakma.

**Formal analysis:** Shopnil Akash, Md. A. K. Azad, Unesco Chakma, Md. Rezaul Islam, Nobendu Mukerjee.

**Investigation:** Shopnil Akash, Farjana Islam Aovi, Md. A. K. Azad, Unesco Chakma, Nobendu Mukerjee, Md. Mominur Rahman, Imren Bayıl, Summya Rashid.

**Methodology:** Shopnil Akash, Md. A. K. Azad, Ajoy Kumer, Md. Mominur Rahman, Imren Bayıl.

**Resources:** Farjana Islam Aovi, Ajoy Kumer, Md. Rezaul Islam, Nobendu Mukerjee.

**Software:** Unesco Chakma, Md. Rezaul Islam.

**Supervision:** Rohit Sharma.

**Validation:** Shopnil Akash, Ajoy Kumer, Unesco Chakma, Md. Rezaul Islam, Md. Mominur Rahman, Imren Bayıl, Summya Rashid.

**Visualization:** Md. A. K. Azad, Md. Rezaul Islam, Md. Mominur Rahman, Imren Bayıl, Summya Rashid.

**Writing – original draft:** Shopnil Akash, Farjana Islam Aovi, Ajoy Kumer.

**Writing – review & editing:** Nobendu Mukerjee, Rohit Sharma.

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
