## [Decision Letter · Decision Letter 0]

3 Apr 2023

PONE-D-23-06412Computational investigation of Scutellarein derivatives as an inhibitor against triple-negative breast cancer by Quantum calculation, and drug-designed approachesPLOS ONE

Dear Dr. Akash,

Thank you for submitting your manuscript to PLOS ONE. After careful consideration, we feel that it has merit but does not fully meet PLOS ONE’s publication criteria as it currently stands. Therefore, we invite you to submit a revised version of the manuscript that addresses the points raised during the review process.

We look forward to receiving your revised manuscript.

Kind regards,

Sheikh Arslan Sehgal, PhD

Academic Editor

PLOS ONE

Journal Requirements:

   "No funding was received in this study."

   "The authors declare that the research was conducted in the absence of any commercial or financial relationships that could be construed as a potential conflict of interest."

Reviewers' comments:

Reviewer's Responses to Questions

**Comments to the Author**

1. Is the manuscript technically sound, and do the data support the conclusions?

Reviewer #1: Yes

Reviewer #2: Yes

2. Has the statistical analysis been performed appropriately and rigorously? 

Reviewer #1: Yes

Reviewer #2: Yes

3. Have the authors made all data underlying the findings in their manuscript fully available?

Reviewer #1: Yes

Reviewer #2: Yes

4. Is the manuscript presented in an intelligible fashion and written in standard English?

Reviewer #1: Yes

Reviewer #2: No

5. Review Comments to the Author

Reviewer #1: The methodology and results obtained are original BUT need to be thoroughly reviewed. Overall, the manuscript is fine, however the discussion is not robust and there are a lot of grammatical issues, spelling mistakes, choice of words and poor result presentation. The authors should provide a valid response(s) to the above comment(s). The details comments are mentioned below:

1. The title should be rewrite, and revised.

2. In 2.1, all the equation should be rechecked, and should be mentioned which software has been used for optimization.

3. In 2.4, Method for Molecular Docking section, it is not clear How did author minimize the energy of the chosen receptors? Why is the removal of water and other unwanted substances necessary during protein preparation?

4. State clearly, how the active sites of the target proteins were determined and total number of active sites like hydrogen bonding, and hydrophobic bonding should be analyzed.

5. In Figure 3. Chemical structure of Scutellarein derivatives is not properly arranged, like 01,02,03,04,05,06,09, 07, 08, 09, and 10, should be rearranged, and corrected the sequences.

6. In labeling of Molecules, author sometimes used DM, sometimes use only number. It should be similar for all compounds. Please check carefully.

7. In table 6, Author have already mentioned standard drugs Capecitabine, but why should you have mentioned Fluconazole? Since it as an antifungal drug. Please check this carefully.

8. The result of molecular docking and interactions are not comprehensively discussed. Please consider revising. Also, there are some identified spelling mistakes. Please correct this appropriately

9. There are a lot of grammatical issues, spelling mistakes and choice of words in the introductory section of this manuscript. Kindly edit and correct appropriately. Use Grammarly software on other available.

I hereby recommend this paper for publication only IF the authors carefully address the comments:

Reviewer #2: The authors have chosen an area that needs attention and has tried to focus on it but there are a lot of lacunas in the study and writing of the manuscript.

1. Why was Scutellarin considered a promising candidate for this study?

2. In the abstract, describe the full name of HOMO and LUMO, since the terms were initially in the MS, after which the short form can be used.

3. What is the relevance of the pass prediction section?

4. Interaction pictures with the selected enzymes should be given. and interactions should be explained one by one.

5. In keyword sections, ‘Molecular Dynamic’ should be ‘Molecular Dynamics Simulation

6. Throughout the manuscript, certain terms are written in capital letters; please replace them with small letters.

7. Mentioned the PyRx and Discovery Studio software versions used for this study.

8. It is important for authors to identify the specific amino acid residues (hydrogen bonding, and hydrophobic bonding) formed during drug-protein complex formation.

9. Suggested updating the quality, or resolution of the images of Molecular docking pocket, hydrogen bonding, and binding interaction.

10. Determination of the Data of ADMET’ should be ‘ADMET profile prediction’

11. Finally, the Conclusion should be rewritten more briefly and covered all the information discussed in the result discussion.

12. On page no. 23 and 24, mention types of interactions with a receptor in figure legends

13. Authors only performed RMSD and RMSF, but these are not sufficient to prove the stability. It is highly recommended to calculate protein-ligand contacts over a period of simulation, radius of gyration, and solvent-accessible surface area of protein.

6. PLOS authors have the option to publish the peer review history of their article (what does this mean?). If published, this will include your full peer review and any attached files.

Reviewer #1: **Yes: **Rana Adnan Tahir (Ph.D.)

Department of Biosciences,

COMSATS University Islamabad, Sahiwal Campus

Reviewer #2: **Yes: **Muhammad Nasir Iqbal

---

## [Author Response · Author response to Decision Letter 0]

5 May 2023

Thank you very much for your letter with reviewer comments. We are pleased to know that our manuscript was rated as potentially admissible for publication in the PLoS one, subject to adequate revision according to reviewer instructions. 

We have completed all corrections and revised the manuscript based on the comments made by the reviewers point by point. So, I am sending herewith our revised manuscript with related documents in your favor. We would like to thank you for allowing us to resubmit a revised version of the manuscript. Please inform if any inconvenience or any other suggestions from your end. 

I would be highly glad if you take the necessary actions regarding our revised manuscript to publish in your PLoS one. 

1. Please ensure that your manuscript meets PLOS ONE's style requirements.

 Response: Thanks for your valuable suggestion. We have revised accordingly.

 "No funding was received in this study."

Response: This is a self-funded project. So, not applicable this section. 

Response: Earlier we have mentioned that no funding was received in this study and mentioned all author of this manuscript has contribution for this research. So, not applicable this section. 

Response: No author received a salary from any of your funders. So, not applicable this section.

Response: The authors received no specific funding for this work.

3.Thank you for stating the following in the Competing Interests section: 

 "The authors declare that no conflict of interest."

Response: We have mentioned in revised cover letter as per as suggestion.

Response: Yes, all data are available to corresponding author, and no restriction for shearing. 

Response: We have mentioned in revised cover letter as per as suggestion.

Response: Yes, all data are available 

Response: No restriction 

Response: Revised all the references accordingly.

Authors Responses to Questions

Reviewer #1: 

1. The title should be rewrite, and revised.

Response: Thanks for your valuable suggestion. We have rewritten the title in revised manuscript.

2. In 2.1, all the equation should be rechecked, and should be mentioned which software has been used for optimization.

Response: Thanks for your valuable suggestion. We have rechecked the equation and Detail's optimization description has been given as per as suggestion.

3. In 2.4, Method for Molecular Docking section, it is not clear How did author minimize the energy of the chosen receptors? Why is the removal of water and other unwanted substances necessary during protein preparation?

Response: Thanks for your valuable comments. We have revised as per as suggestion. The energy of the chosen receptor was minimized using swisspdbviwer application. Besides, to bind a drug in a specific side, like as lock and key model, must be cleaned or fresh target receptor, excess molecules such as water and other unwanted substances may interfere to bind specific site. So, the water and other unwanted substances are cleaned before docking.

4. State clearly, how the active sites of the target proteins were determined and total number of active sites like hydrogen bonding, and hydrophobic bonding should be analyzed.

Response: Thanks for your comments. Author has used discovery studio to visualize or determine active site after molecular docking. The total number of hydrogen bonding, and hydrophobic bonding are given in supplementary table S1. 

5. In Figure 3. Chemical structure of Scutellarein derivatives is not properly arranged, like 01,02,03,04,05,06,09, 07, 08, 09, and 10, should be rearranged, and corrected the sequences.

Response: Thanks for your comments. We have rechecked and corrected as per as suggestion.

6. In labeling of Molecules, author sometimes used DM, sometimes use only number. It should be similar for all compounds. Please check carefully.

Response: Thanks, pointing our mistakes. We have corrected and make the label similar for all compounds.

7. In table 6, Author have already mentioned standard drugs Capecitabine, but why should you have mentioned Fluconazole? Since it as an antifungal drug. Please check this carefully.

Response: Thanks, pointing our mistakes. We removed it in revised manuscript.

8. The result of molecular docking and interactions are not comprehensively discussed. Please consider revising. Also, there are some identified spelling mistakes. Please correct this appropriately

Response: We have discussed clearly and corrected as per as suggestion.

9. There are a lot of grammatical issues, spelling mistakes and choice of words in the introductory section of this manuscript. Kindly edit and correct appropriately. Use Grammarly software on other available.

Response: We have revised and corrected as per as suggestion.

I hereby recommend this paper for publication only IF the authors carefully address the comments:

Reviewer #2: 

1. Why was Scutellarin considered a promising candidate for this study?

Response: Scutellarin has a history of use in innovative medicine containing this compound. Scutellarin may stimulate different types of cancer cells to undergo apoptosis when tested in vitro such as Colorectal cancer Prostate cancer Breast cancer Lung cancer and Renal cell carcinoma. So, based on literatures studies, the Scutellarin has picked up and modified its structure to get more potent drug candidate. Described in introduction section.

2. In the abstract, describe the full name of HOMO and LUMO, since the terms were initially in the MS, after which the short form can be used.

Response: We have given full name of HOMO and LUMO as per as suggestion.

3. What is the relevance of the pass prediction section?

Response: The pass prediction section in in silico drug design refers to the process of predicting whether a potential drug molecule will pass certain criteria. These filters are based on probability of being active (Pa) and probability of being inactive (Pi). By using pass prediction tools and filters, researchers can quickly and efficiently screen large numbers of potential drug candidates, identifying those with the greatest likelihood of success before investing time and resources in their development. This can greatly accelerate the drug discovery process and increase the chances of developing effective treatments for a range of diseases. Details described earlier in manuscript.

4. Interaction pictures with the selected enzymes should be given. and interactions should be explained one by one.

Response: Thanks for your suggestion. We have explained and total number of active sites has given in Fig 7 and explanation is also given in section 3.10.

5. In keyword sections, ‘Molecular Dynamic’ should be ‘Molecular Dynamics Simulation

Response: Thanks, pointing our mistakes. We made correction in revised manuscript.

6. Throughout the manuscript, certain terms are written in capital letters; please replace them with small letters.

Response: Thanks, pointing our mistakes. We have revised.

7. Mentioned the PyRx and Discovery Studio software versions used for this study.

Response: We have updated in revised manuscript.

8. It is important for authors to identify the specific amino acid residues (hydrogen bonding, and hydrophobic bonding) formed during drug-protein complex formation.

Response: The total number of hydrogen bonding, and hydrophobic bonding are given in supplementary Table S1. 

9. Suggested updating the quality, or resolution of the images of Molecular docking pocket, hydrogen bonding, and binding interaction.

Response: Thanks for your advised. We have increased the resolution up to 600 DPI in revised manuscript

10. Determination of the Data of ADMET’ should be ‘ADMET profile prediction’

Response: We have revised and replaced.

11. Finally, the Conclusion should be rewritten more briefly and covered all the information discussed in the result discussion.

Response: Revised as per as suggestion.

12. On page no. 23 and 24, mention types of interactions with a receptor in figure legends

Response: Revised as per as suggestion.

13. Authors only performed RMSD and RMSF, but these are not sufficient to prove the stability. It is highly recommended to calculate protein-ligand contacts over a period of simulation, radius of gyration, and solvent-accessible surface area of protein.

Response: Thanks for your advised. We have calculated protein-ligand contacts over a period of simulation, radius of gyration, and solvent-accessible surface area of protein in section 3.11.

---

## [Editor Report · Decision Letter 1]

7 May 2023

A drug design strategy based on molecular docking and molecular dynamics simulations applied to development of inhibitor against triple-negative breast cancer by Scutellarein derivatives

PONE-D-23-06412R1

Dear Dr. Akash,

We’re pleased to inform you that your manuscript has been judged scientifically suitable for publication and will be formally accepted for publication once it meets all outstanding technical requirements.

Kind regards,

Sheikh Arslan Sehgal, PhD

Academic Editor

PLOS ONE
---

## [Editor Report · Acceptance letter]

5 Jun 2023

PONE-D-23-06412R1 

A drug design strategy based on molecular docking and molecular dynamics simulations applied to development of inhibitor against triple-negative breast cancer by Scutellarein derivatives 

Dear Dr. Akash:

I'm pleased to inform you that your manuscript has been deemed suitable for publication in PLOS ONE. Congratulations! Your manuscript is now with our production department. 

Kind regards, 

on behalf of

Dr Sheikh Arslan Sehgal 

Academic Editor

PLOS ONE